# Comparison between Blood, Non-Blood Fluids and Tissue Specimens for the Analysis of Cannabinoid Metabolites in Cannabis-Related Post-Mortem Cases

Torki A. Zughaibi [1,2,*,†], Latifa Al-Qumsani [1,2,†], Ahmed A. Mirza [1,2], Amal Almostady [2], Jude Basrawi [2], Shams Tabrez [1,2], Faiz Alsolami [3], Rami Al-Makki [3], Sami Al-Ghamdi [3], Abdullah Al-Ghamdi [3], Abdulnasser E. Alzahrani [3], Majda Altowairqi [3], Hassan Alharbi [3], Michelle R. Peace [4], Majed A. Halwani [5] and Ahmed I. Al-Asmari [6,*]

1   Department of Medical Laboratory Sciences, Faculty of Applied Medical Sciences, King Abdulaziz University, Jeddah 21589, Saudi Arabia
2   King Fahd Medical Research Center, King Abdulaziz University, Jeddah 21589, Saudi Arabia
3   Poison Control and Forensic Chemistry Center, Ministry of Health, Jeddah 21176, Saudi Arabia
4   Department of Forensic Science, Virginia Commonwealth University, 1015 Floyd Avenue, Richmond, VA 23284, USA
5   Nanomedicine Department, King Abdullah International Medical Research Center, King Saud bin Abdulaziz University for Health Sciences, Riyadh 14611, Saudi Arabia
6   King Abdul-Aziz Hospital, Ministry of Health, Jeddah 22421, Saudi Arabia
*   Correspondence: taalzughaibi@kau.edu.sa (T.A.Z.); ahmadalasmari@yahoo.com (A.I.A.-A.)
†   These authors contributed equally to this work.

**Abstract:** Cannabis use is widespread and is one of the most common drugs encountered in forensic-related analysis (antemortem and postmortem cases). However, the correlation between illicit cannabis use and death is rarely investigated, even while taking into consideration its role in the central nervous system depression and cardiovascular disorders. Few studies have discussed other non-blood specimens; this has brought a special interest in analyzing THC and its metabolites in different body parts in order to make precise forensic decisions. Herein, we are investigating the presence of $\Delta^9$-tetrahydrocannabinol (THC) and its metabolites:(11-hydroxy-$\Delta^9$-tetrahydrocannabinol (THC-OH) and 11-nor-$\Delta^9$- tetrahydrocannabinol-9-carboxy (THC-COOH)) in different postmortem specimens. Forty-three cases of bodily fluids and tissue post-mortem samples, previously found to be cannabinoid-positive were analyzed in the current investigation using alkaline hydrolysis followed by solid phase extraction and LC-MS/MS for THC and its metabolites concentration. In the current study, the highest median THC-COOH and THC-OH concentrations were detected in bile samples (1380 ng/mL and 8 ng/mL, respectively), while the highest THC median concentration was detected in gastric contents (48 ng/mL). This can be explained due to the postmortem distribution of blood to other bodily fluids and tissues and the accumulation in bile following multiple doses. Furthermore, high THC levels in gastric contents can be explained by the undergoing cycles of entero-hepatic circulation which resulted in a significant increase in THC in gastric contents. THC-COOH can be the best indicator to detect cannabinoids in toxicology studies, thus the inclusion of active THC metabolites is essential in death investigations. Additionally, THC-OH concentrations in postmortem cases could be influenced by body mass index. In this study, all types were specimens found to be suitable for testing cannabinoid metabolites, except for vitreous humor which showed low rates of detectability for cannabinoid metabolites.

**Keywords:** cannabinoids; forensic toxicology; LC-MS/MS; post-mortem analysis

## 1. Introduction

In a 2017 United Nations' World Drug Report, it was shown that about 271 million people consumed prohibited drugs with cannabis alone accounting for about 188 million

cases, making it the most used drug worldwide. For instance, in the United States, the number of cannabis users increased by 60% between 2007 and 2017 [1]. In fact, many countries legalized medical and recreational cannabis use due to its therapeutic benefits. In consequence, increased availability and use of cannabis has led to an increase in cannabinoid-related incidents encountered in forensic analysis such as increased emergency department visits, children's ingestion of edible food containing cannabinoids, and cannabis-induced impaired driving [2,3]. In the Middle East, especially in Arab countries (except Lebanon), cannabis is still not legalized and is considered a drug of abuse [4]. The impact of cannabis use and its role in postmortem-related cases in these countries is rarely reported. In a recent study from Saudi Arabia, cannabinoids were the 4th most detected drug in postmortem populations reported between 2016 and 2018 in the city of Jeddah [5]. The use of cannabis can result in death and would require forensic postmortem investigations. Moreover, the increased risk of cardiovascular toxicity and sudden death was reported [6,7]. Therefore, it is an important task for forensic toxicologists to detect cannabinoids and their metabolites [8,9].

The main psychoactive compound of cannabis is Δ9-tetrahydrocannabinol (THC), which produces various pharmacologic effects in humans and animals [10]. Following ingestion in the body, THC is rapidly metabolized into 11-hydroxy-Δ9-tetrahydrocannabinol (11-OH-THC), which then further metabolizes to the inactive form, 11-nor—Δ9-tetrahydrocannabinol-9-carboxy (THC-COOH) [11]. Numerous measurement methods can be used to analyze THC, 11-OH-THC, and THC-COOH in human postmortem specimens. However, the gas chromatography coupled mass spectrometry (GC-MS) [11–13] and high-performance liquid chromatography integrated with a single quadrupole mass spectrometer (LC-MS), or tandem mass spectrometry (LC-MS/MS) techniques were the most commonly used [11,13–15]. In addition, LC-MS/MS techniques have more advantages when compared to standard GC-MS methods. Due to its simple nature and non-derivative sample preparation, it is highly specific and sensitive and represents the most effective method for analyzing various cannabinoids and their corresponding metabolites [13,14].

The differences in metabolite concentrations in the previous studies varied between individual subjects in many ways, such as weight, tolerance, frequency of usage, possible polydrug intoxication, stability of the drug's metabolites in postmortem tissues, and duration between drug intake and death (PMI). The average duration for detecting the prevalent metabolite, THC-COOH, in plasma samples is typically within the range of 2–7 days, depending on the dosage. However, it is possible for THC-COOH to persist in the bloodstream for weeks after being consumed [16]. As opposed to the previously mentioned THC-COOH, THC and THC-OH peak concentration in the blood can be achieved within a span of 15–20 min after administration. Both metabolites have short half-lives and as a result are distributed to other tissues within a few minutes of cannabinoid intake [17]. In a previous study, a higher concentration of metabolites (THC, THC-OH, and THC-COOH) was found in blood collected in blood collection tubes that had no sodium fluoride [9]. In the same study, the ratio between the concentrations of (THC, THC-OH, and THC-COOH) detected in blood tubes containing sodium fluoride/without sodium fluoride were 93%, 86% and 78%, correspondingly. Despite the low levels of THC-OH observed in the study, its analysis is crucial as it serves as a temporary metabolite of THC and can indicate recent cannabinoid use among cases involved in grisly incidents [11].

Urine and blood are the most frequently used biological samples in forensic toxicological drug analysis. This is due to the fact that they are readily available in most cases, are homogenous in nature, and have ready-to-run applicability, especially in modern LC-MS technology. While blood and urine samples are preferred for post-mortem toxicology analysis of cannabis, they may not always be available for testing due to accident-related fatalities leading to loss of blood or other circumstances such as decomposition or exsanguination [3,11,15].

For cases where blood samples are not available for postmortem analysis, reliable non-blood post-mortem samples are critical to explore as substitute samples for the quantifi-

cation and identification of THC and its metabolites. Hence, our objective is to investigate the distribution of THC concentrations in different postmortem bodily and solid tissue specimens. This was done by evaluating concentrations of THC, THC-OH and THC-COOH in post-mortem specimens including blood, urine, liver, vitreous humor, kidney, bile, gastric contents, and brain using solid phase extraction (SPE). These analytes were then measured using LC-MS/MS. The concentration of analytes of interest was correlated with PMI, age, and body mass index (BMI). The aim of this study was to assess and highlight the importance of non-blood specimens; our previously published reports were used as a reference to compare the concentrations and outcomes with the results obtained from blood, gastric contents, liver, and kidney samples.

## 2. Materials and Methods

### 2.1. Materials

The standards and internal standards, namely THC, THC-OH, THC-COOH, THC-d3, THC-OH-d3, and THC-COOH-d9 obtained through Lipomed AG, Arlesheim, Switzerland. The HPLC grade solvents such as methanol (99.8%), ethyl acetate, acetonitrile, sodium hydroxide, concentrated hydrochloric acid (HCL), hexane and ammonium formate were obtained from Sigma Aldrich, Merck, KGaA. The cartridges were Clean Screen® SPE (CSTHCU203. 200 mg/3 mL), UCT Bristol, PA, USA.

### 2.2. Case Study

#### 2.2.1. Study Design

This study was a cross-sectional examination of the concentration distribution of THC and its metabolites in various bodily fluids and tissue specimens from autopsies. The 43 cases in the study, which all tested positive for cannabinoids, were referred to the Jeddah Poison Control and Forensic Medical Chemistry Centre (JPCC) in Jeddah, Saudi Arabia and had received ethical approval (no. H-02-J-002, Ethical Approval Committee, Ministry of Health, Jeddah Health Affairs). Blood and other desired samples (urine, liver, vitreous humor, kidney, bile, gastric contents, and brain) were collected and preserved at -20 °C for analysis. The deceased individuals' data, including sex, age, weight, history, and cause of death, were obtained from the Forensic Toxicology Reports Database (FTRJ), an online system that stores information on forensic medical cases.

#### 2.2.2. Specimens Collection

Forensic pathologists at Jeddah Forensic Medicine departments collected multiple specimens to investigate the distribution of THC and its metabolites. Blood with sodium fluoride (BNaF) samples were obtained from subclavian sites and contained 1% sodium fluoride. In order to avoid contamination, samples were collected from 3 sites across the deep right lobe of the liver. Samples were collected from both kidneys, specifically from the center of the left and right kidney organs. Only 1 cm$^3$ of central brain tissues was collected for analysis from at least three sites for analysis. Vitreous humor (VH) samples were obtained from both eyes and combined in a gray test tube; vitreous humor was extracted from both eyes using [18]. Preservative urine samples were directly obtained from the bladder. All available gastric contents at the time of autopsy were retained and used for analysis. Specimens collected during the autopsy included BNaF in 39 cases (69%), vitreous humor in 31 cases (72%), urine in 37 cases (86%), gastric contents in 19 cases (44%), liver tissue in 15 cases (34%), kidney tissue in 13 cases (30%), bile in 17 cases (40%), and brain tissues in 4 samples (9%).

### 2.3. Sample Preparation

#### 2.3.1. Non-Hydrolyzed Specimens

One milliliter of bodily fluids (urine, bile, and gastric contents) included in this study was subjected to alkaline hydrolysis. Internal standards (50 ng/mL) of (THC-d3, THC-OH-d3, and THC-COOH-d9) were added. 200 µL of sodium hydroxide (10 N) was added to

each specimen except BNaF and VH samples. All samples were further kept for 20 min in a 60 °C water bath before proceeding with the hydrolysis. Glacial acetic acid (2 mL) was added to adjust the sample's pH to 3.5. For solid tissues (liver, kidneys, and brain) a gram of the selected tissue specimen was placed into a stomacher bag for 5 min after diluting 2:1 (aqueous 1% sodium fluoride: tissue) followed by homogenization in the stomacher (Seward Limited; West Sussex, UK). Homogenate tissue (0.5 g) was then transferred to a 15.0 mL glass tube. 50 ng/g of internal standard was added to all samples. Next, 200 µL of sodium hydroxide (10 N) was added to all specimens and incubated for 20 min at 60 °C. After cooling the samples for 5 min, glacial acetic acid (2 mL) was added to adjust the pH to 3.5. Tubes were subjected to centrifugation for 10 min at $2200 \times g$. The supernatants were then placed in clean test tubes.

### 2.3.2. Solid Phase Extraction (SPE)

All specimens were extracted using labeled clean screen® cartridges as described in Al-Asmari report [9,14]. Briefly, SPE cartridges were placed on the vacuum manifold. 3 mL of methanol was added and allowed to pass through it, then 3 mL of deionized water was added before adding 1 mL of HCl (0.1 M). The specimens were then allowed to pass through the labeled cartridge completely using gravity. Then, 2 mL of deionized water was added and allowed to flow through before adding HCl (0.1 M): acetonitrile (70:30). Next, the SPE cartridges were dried through full vacuum for 5 min at (>10 inches Hg) before adding 200 µL of hexane. Two mL of hexane and ethyl acetate (50:50) were used for elution. In order to inject the samples (1 µL) inside the LC-MS/MS, eluents were dried at 40 °C for 20 min before reconstitution by the initial mobile phase (100 µL).

### 2.4. LC–MS/MS Conditions

In the current investigation, the authors utilized a LC-MS/MS method previously described in the literature [9] for the detection and quantification of the target analytes. The Shimadzu LCMS-8050 triple quadrupole mass spectrometer coupled with the Nexera UHPLC system (Kyoto, Japan) was employed. The Raptor Biphenyl column (50 × 3.0 mm, 2.7 µm) in combination with the Security Guard column (Raptor Biphenyl, 2.7 µm, 5/3.0 mm; Restek, USA) were maintained at a constant temperature of 4 °C and 40 °C, respectively. The analytes of interest were separated using a gradient elution with a mobile phase containing ammonium formate (10 mM), pH 3 (A) and methanol (B) at a flow rate of 0.3 mL/min. The gradient elution was conducted with 70% of solution B for one min as the initial mobile phase elution. This was then gradually increased to 95% for solution B up to minute 5 and then maintained for 3 min. At minute nine it was returned to 70% before the last minute of the next injection. In this experiment, an electrospray ionization source (ESI) was used. This assay was conducted by Multiple Reaction Monitoring (MRM) positive ion mode. Details for analytes of THC, its metabolites and their respective internal standards as produced by LC-MS/MS are shown in Table 1.

### 2.5. Method Validation

The forensic toxicology international guidelines [19,20] were followed to validate the experimental procedures used in this study. The SPE and LC-MS/MS methods, previously published in [9,10], were used to extract the analytes of interest. The THC and its metabolites' calibration curves were linear with coefficients of determination greater than 0.99 and covered the range of 1.0–1000 ng/mL for bodily fluids specimens and 1.0–1000.0 ng/g for tissue specimens. The LOD ranged from 0.5 to 1.3 ng/mL and 0.2–0.8 ng/mL, while the LOQ ranged from 1.0 to 2.0 ng/mL and 1.0 ng/g for bodily fluids and tissue specimens, respectively (Table S1 and Figure 1). The precision of the within-run and between-run was less than 11%. Three controls (25 ng/mL, 100 ng/mL, and 750 ng/mL) were used to test precision, and the accuracy values ranged from −8% to +8%. The matrix effects of THC and its metabolites were determined using three controls (25 ng/mL, 100 ng/mL, and 750 ng/mL) and ranged from 78.0% to 122%, while the analytical recoveries ranged

from 79% to 97%. The dilution controls fell within the acceptable range ($\pm15\%$) for method validation, indicating that the assay was reliable. Moreover, in this study, no interference was detected from blank postmortem specimens, commonly encountered compounds, or carryover effects from the previous injection. Method validation results are detailed in Table S2.

**Table 1.** Liquid chromatography tandem mass spectrometry parameters for the analysis of Δ9—tetrahydrocannabinol and its metabolites.

| | Parameters | Δ9—Tetrahydrocannabinol (THC) | 11-nor-Δ9-THC-9-Carboxy Acid | 11-Hydroxy-Δ9-THC |
|---|---|---|---|---|
| Analytes | Precursor Ion ($m/z$) | 315 | 345 | 331 |
| | Product Ion(s) ($m/z$) | 193, 123, 93 | 193, 299, 119 | 193, 201, 123 |
| | Quantifier Ion ($m/z$) | 315.0→123 | 345→193 | 331→193 |
| | Qualifier Ion ($m/z$) | 315→193 | 345→123 | 331→201 |
| | Reference ion ratios | 63 | 70 | 69 |
| | Retention time (Min) | 5.7 | 5.0 | 4.8 |
| Internal standards | Internal Standard (IS) | THC-d$_3$ | THC-COOH-d$_9$ | THC-OH-d$_3$ |
| | IS Precursor Ion(s) ($m/z$) | 318 | 354 | 334 |
| | Product Ion ($m/z$) | 198, 123, 107 | 197, 123, 308 | 196, 201, 105 |
| | Quantifier Ion ($m/z$) | 318→123 | 354→197 | 334→196 |
| Analytes and their internal standards | Dwell time (msec) | 10 | 10 | 10 |
| | Q1 Bias (V) | −15 | −15 | −15 |
| | Collision energy (%) | −35 | −35 | −35 |
| | Q3 Bias (V) | −15 | −15 | −15 |
| | Nebulizing Gas Flow (L/Min) | 3 | 3 | 3 |
| | Interface Temperature °C | 300 | 300 | 300 |
| | Heater Block Temperature °C | 400 | 400 | 400 |

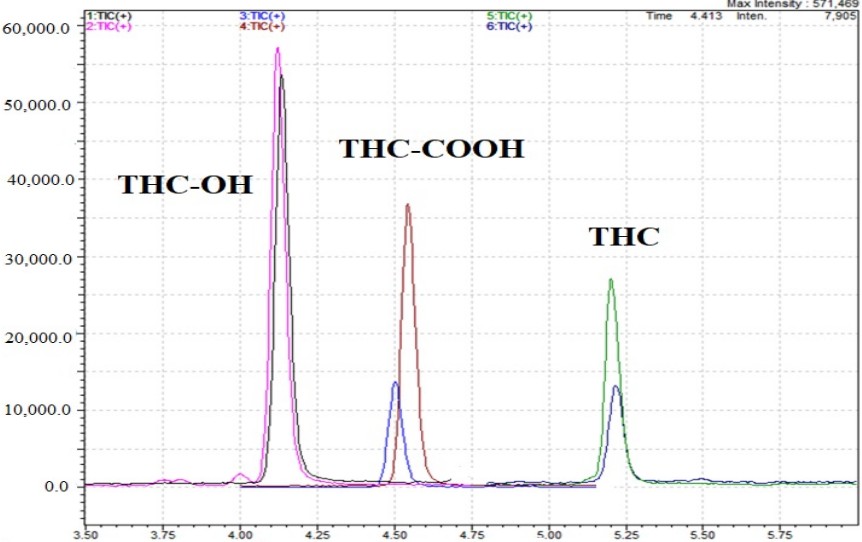

**Figure 1.** Blank blood with sodium fluoride fortified Δ9-Tetrahydrocannabinol (THC), 11-nor—Δ9-Tetrahydrocannabinol -9-Carboxy acid (THC-COOH) and 11-Hydroxy-Δ9- Tetrahydrocannabinol (THC-OH) at 1 ng/mL.

## 3. Results

### 3.1. Demographic Profile (Table 2)

**Table 2.** Baseline characteristics of the study patients (*n* = 43).

| Study Data | N (%) |
|---|---|
| Age in years (mean ± SD) | 32.1 ± 11.9 |
| Gender | |
| Male | 40 (93.0%) |
| Female | 03 (07.0%) |
| BMI level | |
| Normal (18.5–24.9 kg/m$^2$) | 20 (46.5%) |
| Overweight (25–29.9 kg/m$^2$) | 16 (37.2%) |
| Obese (≥30 kg/m$^2$) | 07 (16.3%) |
| Previous history of drug abuse | |
| Yes | 08 (18.6%) |
| No | 35 (81.4%) |
| Use of other drugs | |
| Yes | 36 (83.7%) |
| No | 07 (16.3%) |

### 3.2. Multiple Specimens Analysis

Concentrations of cannabinoids were determined in postmortem fluids and tissues from 43 postmortem cases included in this work (Table S3). THC or its metabolites were quantified in at least one specimen. In the BNaF: THC, THC-COOH and THC-OH were confirmed in 33 cases (range: 1–20 ng/mL), 38 cases (range: 1–50 ng/mL), and 30 out of 38 cases (range: 1–3 ng/mL) respectively. In urine samples, THC, THC-COOH and THC-OH were found to be positive in 28 out of 37 cases (range: 1–51 ng/mL), 37 cases (range: 5–3700 ng/mL), and 21 cases (range: 0.6–2 ng/mL) respectively. Moreover, THC, THC-COOH and THC-OH were identified in vitreous humor in fewer samples, THC was detected in 6 out of 17 cases (median: 2 ng/mL, range: 1–2 ng/mL), THC-COOH was detected in five cases (median:1.0 ng/mL, range: trace concentration-1.0 ng/mL), while THC-OH was found in only three cases with concentrations ranging from trace concentrations to 1.0 ng/mL. THC, THC-COOH and THC-OH were detected in liver and kidney tissues in fewer samples (12 and 11 cases respectively). The THC-COOH concentration was found to have a median concentration of 42 ng/g and 15 ng/g in the two matrices, while THC and THC-OH concentrations were low with median concentrations of 2 ng/g and 1 ng/g for THC and THC-OH in both matrices, respectively. All 17 bile specimens were found to be positive for THC-COOH (median: 1500 ng/mL, range: 220–46200 ng/mL). Out of 17 cases, 12 were positive for THC (median: 34 ng/mL, range: 9–247 ng/mL), and 13 were positive for THC-OH (median: 10 ng/mL, range: 1–22 ng/mL). In gastric contents, THC, THC-COOH and THC-OH were confirmed to be positive in 15, 11, and 7 out of 15 cases (median: 56 ng/mL, range: 9–310 ng/mL), (median: 12 ng/mL range: 2–100 ng/mL) and (median: 2.5 ng/mL, range: 2–35 ng/mL). In the four brain tissue samples available for testing, THC, THC-COOH and THC-OH were detected with mean concentrations of (range: trace level to 3.0 ng/g), (range: 1 to 2.2 ng/g) and (range: 1 to 1.5 ng/g), respectively. The mean concentrations of analytes of interest in all matrices included in this study are summarized in Table 3.

### 3.3. THC in Different Specimens

Table 4 displays the results of the correlation analysis. The study found a strong positive correlation (rs = 0.444; *p* = 0.007) between THC metabolites in blood and urine. Additionally, a positive correlation was observed between liver and blood (rs = 0.709; *p* = 0.015). Blood showed significant positive correlations with liver (rs = 0.683; *p* = 0.042), urine (rs = 0.461; *p* = 0.010), and VH (rs = 0.401; *p* = 0.026). Furthermore, statistically significant positive correlations were found between kidney and urine (rs = 0.765; *p* = 0.016)

as well as liver (rs = 0.828; *p* < 0.001). However, all other correlations were not statistically significant. A highly positive correlation between THC metabolites was found in blood and urine with (rs = 0.444; *p* = 0.007). Liver and VH samples also showed a positive correlation (rs = 0.683; *p* = 0.042). Furthermore, BNaF showed positive significant correlations with liver (rs = 0.710; *p* = 0.015), urine (rs = 0.461; *p*= 0.010) and VH (rs = 0.401; *p* = 0.026). Additionally, there was a positive statistically significant correlation between kidney and urine (rs = 0.765; *p* = 0.016) and liver (rs = 0.828; *p* < 0.001).

**Table 3.** Descriptive statistics of THC and its metabolites.

| Variables | Δ9—Tetrahydrocannabinol (ng/mL)/(ng/g) Mean (95% CI) | 11-nor—Δ9-THC-9-Carboxy Acid (ng/mL)/(ng/g) Mean (95% CI) | 11-Hydroxy-Δ9-THC (ng/mL)/(ng/g) Mean (95% CI) |
|---|---|---|---|
| Blood | 13.1 (−9.2–33.8) [a] | 17.1 (6.48–29.5) | 1.61 (−0.07–1.43) |
| Urine | 4.39 (−2.5–19.6) [a] | 383.6 (27.7–765) [a] | 0.62 (0.02–1.11) |
| Liver | 1.71 (0.88–2.47) | 80.6 (−24.7–199) [a] | 0.79 (0.04–1.27) |
| VH | 0.98 (−0.02–2.04) | 0.89 (−0.02–2.04) | 0.46 (−0.05–1.16) |
| Kidney | 3.39 (−2.66–13.7) | 118.8 (−2.66–13.7) | 0.63 (0.22–1.32) |
| Bile | 106.1 (−14.3–226) [a] | 8808.6 (1684 15932) [a] | 9.47 (4.97–13.9) |
| Gastric Contents | 80.9 (−41.7–121) | 68.3 (−41.7–121) | 3.51 (−1.06–4.96) |
| Brain | 2.02 (0.21–3.83) | 1.15 (−0.65–1.66) | 0.72 (−0.02–1.46) [a] |

[a]: Non-normally distributed data.

**Table 4.** Correlation (Spearman-Rho) between Δ9—Tetrahydrocannabinol in different bodily and tissue specimens.

| Specimens | Blood | Urine | Liver | VH |
|---|---|---|---|---|
| Blood | 1 | | | |
| Urine | 0.444 ** | 1 | | |
| Liver | 0.709 * | 0.562 | 1 | |
| VH | 0.401 * | 0.461 * | 0.683 * | 1 |
| Kidney | 0.510 | 0.765 * | 0.828 ** | 0.500 |
| Bile | 0.243 | 0.053 | 0.203 | −0.147 |
| Gastric contents | 0.320 | 0.053 | −0.012 | −0.165 |
| Brain | −0.500 | – | −0.500 | – |

* Correlation is significant at 0.05 level (2-tailed). ** Correlation is significant at 0.01 level (2-tailed).

### 3.4. THC-OOOH in Different Specimens

Table 5 presents the results of the correlation analysis conducted in this study. The findings revealed a significant correlation between THC-OOH metabolites in urine and blood (rs = 0.641; *p* < 0.001). Furthermore, highly significant correlations were observed between liver and blood (rs = 0.882; *p* < 0.001), liver and urine (rs = 0.761; *p* < 0.001), kidney and blood (rs = 0.817; *p* < 0.001) and liver (rs = 0.895; *p* < 0.001). Bile also showed a significant correlation with liver (rs = 0.735; *p* = 0.010), while gastric contents had a significant correlation with bile (rs = 0.742; *p* = 0.002). All other correlations in the analysis were found to be non-significant. A significant correlation was found between urine and blood (rs = 0.641; *p* < 0.001) of THC-OOH metabolites. In addition, the correlation between liver and blood (rs = 0.882; *p* < 0.001), liver and urine (rs = 0.761; *p* < 0.001) were also found to be highly significant. Moreover, a highly significant correlation was found between the kidney and blood (rs = 0.817; *p* < 0.001) and liver (rs = 0.895; *p* < 0.001). Whereas bile was found to have a significant correlation with liver (rs = 0.735; *p* = 0.010). Similarly, gastric contents showed a significant correlation with bile (rs = 0.742; *p* = 0.002).

**Table 5.** Correlation (Spearman-Rho) between 11-nor—Δ9-Tetrahydrocannabinol-9-Carboxy acid in different bodily and tissue specimens.

| Specimens | Blood | Urine | Liver | VH |
|---|---|---|---|---|
| Blood | 1 | | | |
| Urine | 0.641 ** | 1 | | |
| Liver | 0.882 ** | 0.761 ** | 1 | |
| VH | 0.182 | −0.014 | 0.517 | 1 |
| Kidney | 0.817 ** | 0.460 | 0.895 ** | 0.214 |
| Bile | 0.404 | 0.216 | 0.735 ** | 0.273 |
| Gastric contents | 0.353 | 0.300 | 0.571 | −0.341 |
| Brain | – | – | 0.500 | – |

** Correlation is significant at 0.01 level (2-tailed).

### 3.5. THC-OH in Different Specimens

A significant correlation was found between blood and urine for each THC metabolite (rs = 0.715; $p < 0.001$). A highly significant correlation was observed between liver and urine (rs = 0.908; $p < 0.001$). Furthermore, the correlation between liver (rs = 0.673; $p = 0.033$), urine (rs = 0.790; $p < 0.001$) and VH among blood (rs = 0.716; $p < 0.001$) were statistically significant. Likewise, there was a significant correlation between the kidney in relation to liver (rs = 0.873; $p < 0.001$) and urine (rs = 0.678; $p = 0.045$). Lastly, the correlation between gastric contents regarding VH (rs = 0.588; $p = 0.021$) and urine (rs = 0.519; $p = 0.033$) was also statistically significant. All the other correlations were non-significant (Table 6).

**Table 6.** Correlation (Spearman-Rho) between 11-Hydroxy-Δ9-Tetrahydrocannabinol in different bodily and tissue specimens.

| Specimens | Blood | Urine | Liver | VH |
|---|---|---|---|---|
| Blood | 1 | | | |
| Urine | 0.715 ** | 1 | | |
| Liver | 0.561 | 0.908 ** | 1 | |
| VH | 0.716 ** | 0.790 ** | 0.673 * | 1 |
| Kidney | 0.357 | 0.678 * | 0.873 ** | 0.565 |
| Bile | 0.423 | 0.424 | 0.014 | 0.509 |
| Gastric contents | 0.388 | 0.519 * | 0.171 | 0.588 * |
| Brain | – | – | 0.500 | – |

* Correlation is significant at 0.05 level (2-tailed); ** Correlation is significant at 0.01 level (2-tailed).

### 3.6. The Role of Other Drugs Detected

Amphetamine was found to be the most common drug detected along with cannabinoids (66.7%), followed by methamphetamine (27.8 %) (Tables 7 and 8 and Figure 2). As can be seen clearly from Table 7 there was no statistically significant correlation between most THC and its metabolites when other drugs were used in combination with cannabinoids as well as when only cannabinoids were used. The use of other drugs in relation to THC metabolites showed statistically significant mean values of THC in the kidneys compared to those who used drugs (T = −5.014; $p < 0.001$). Other THC metabolites were not found to be statistically significant compared to other drugs. Comparing THC-OOH metabolites in relation to other drugs showed a significant association with a lower mean value of THC-OOH in the liver (T = −2.444; $p = 0.033$). Other THC-OOH metabolic factors did not differ significantly compared to the use of other drugs. Finally, when analyzing the relationship between THC-OH metabolic elements and the use of other drugs, there was no significant observation found (all $p > 0.05$).

**Table 7.** Comparison of THC and its metabolites in relation to the use of other drugs.

| | Use of Other Drugs | | *t*-Test | *p*-Value |
|---|---|---|---|---|
| | **Metabolites** | | | |
| **Specimens** | **Yes** <br> **Mean ± SD** | **No** <br> **Mean ± SD** | | |
| | Δ9—Tetrahydrocannabinol | | | |
| Blood [a] | 14.6 ± 49.4 | 4.94 ± 2.01 | 0.473 | 0.091 |
| Urine [a] | 4.29 ± 8.99 | 5.22 ± 3.77 | −0.201 | 0.222 |
| Liver [b] | | 2.11 ± 0.61 | −0.621 | 0.545 |
| VH [b] | 1.61 ± 1.34 | 0.79 ± 0.89 | 0.557 | 0.582 |
| Kidney [b] | 1.81 ± 2.17 | 12.1 ± 5.63 | −5.014 | <0.001 ** |
| Bile [a] | 94.4 ± 236.7 | 293.3 ± 0 | −0.815 | 0.235 |
| Gastric Contents [b] | 68.6 ± 96.8 | 185.9 ± 45.9 | −1.660 | 0.115 |
| | 11-nor—Δ9-Tetrahydrocannabinol-9-Carboxy acid | | | |
| Blood [b] | 15.4 ± 16.1 | 26.9 ± 16.8 | −1.599 | 0.118 |
| Urine [a] | 379.9 ± 686.3 | 416.6 ± 435.7 | −0.095 | 0.620 |
| Liver [a] | 69.6 ± 148.2 | 124.7 ± 75.4 | −0.612 | 0.136 |
| VH [b] | 0.92 ± 0.56 | 0.75 ± 0.55 | 0.622 | 0.539 |
| Kidney [b] | 54.8 ± 148.7 | 471.3 ± 565.3 | −2.444 | 0.033 ** |
| Bile [a] | 8723.4 ± 14305.9 | 10,172.9 ± 0 | −0.098 | 0.588 |
| Gastric Contents [b] | 72.4 ± 237.9 | 31.6 ± 2.21 | 0.237 | 0.816 |
| | 11-Hydroxy-Δ9-Tetrahydrocannabinol | | | |
| Blood [b] | 1.77 ± 5.67 | 0.71 ± 0.73 | 0.452 | 0.654 |
| Urine [b] | 0.63 ± 0.61 | 0.53 ± 0.62 | 0.308 | 0.760 |
| Liver [b] | 0.68 ± 0.51 | 1.28 ± 1.12 | −1.475 | 0.164 |
| VH [b] | 0.48 ± 0.46 | 0.39 ± 0.44 | 0.416 | 0.680 |
| Kidney [b] | 0.65 ± 0.43 | 0.49 ± 0.70 | 0.433 | 0.674 |
| Gastric contents [b] | 3.09 ± 8.09 | 7.24 ± 10.2 | −0.676 | 0.507 |

[a] *p*-value has been calculated using Mann Whitney U-test. [b] *p*-value has been calculated using independent *t*-test; ** Significant at $p < 0.05$ level.

**Table 8.** Comparison of THC and its metabolites in relation to the history of drug abuse.

| Specimens | History of Drug Abuse | | *t*-Test | *p*-Value |
|---|---|---|---|---|
| | **Metabolites** | | | |
| | **Yes** <br> **Mean ± SD** | **No** <br> **Mean ± SD** | | |
| | Δ9—Tetrahydrocannabinol | | | |
| Blood [a] | 5.53 ± 4.88 | 14.7 ± 50.1 | −0.481 | |
| Urine [a] | 4.22 ± 4.43 | 4.44 ± 9.32 | −0.059 | |
| Liver [b] | 1.65 ± 1.16 | 1.72 ± 1.26 | −0.081 | |
| VH [b] | 0.97 ± 0.84 | 0.98 ± 0.80 | −0.019 | |
| Kidney [b] | 1.44 ± 1.00 | 3.76 ± 4.99 | −0.632 | |
| Bile [a] | 83.3 ± 111.1 | 113.1 ± 264.3 | −0.216 | |
| Gastric contents [b] | 30.9 ± 37.6 | 98.8 ± 108.8 | −1.345 | |
| | 11-nor—Δ9-Tetrahydrocannabinol-9-Carboxy acid | | | |
| Blood [b] | 22.2 ± 19.4 | 16.0 ± 15.9 | 0.900 | |
| Urine [a] | 705.3 ± 1325.0 | 308.5 ± 381.4 | 1.456 | |
| Liver [a] | 145.8 ± 125.2 | 64.3 ± 139.1 | 0.922 | |
| VH [b] | 1.06 ± 0.48 | 0.85 ± 0.57 | 0.811 | |
| Kidney [b] | 250.8 ± 352.4 | 94.9 ± 258.6 | 0.756 | |
| Bile [a] | 16,320.2 ± 17,012.6 | 6497.4 ± 12,610.9 | 1.263 | |
| Gastric contents [b] | 235.6 ± 438.5 | 12.6 ± 26.7 | 2.076 | |
| | 11-Hydroxy-Δ9-Tetrahydrocannabinol | | | |
| Blood [b] | 0.89 ± 0.71 | 1.76 ± 0.76 | −0.393 | |
| Urine [b] | 0.77 ± 0.85 | 0.59 ± 0.54 | 0.710 | |

**Table 8.** *Cont.*

| Specimens | History of Drug Abuse | | *t*-Test | *p*-Value |
|---|---|---|---|---|
| | Metabolites | | | |
| | Yes<br>Mean ± SD | No<br>Mean ± SD | | |
| Liver [b] | 1.10 ± 0.88 | 0.72 ± 0.63 | | 0.877 |
| VH [b] | 0.59 ± 0.46 | 0.43 ± 0.45 | | 0.801 |
| Kidney [b] | 0.47 ± 0.62 | 10.2 ± 8.79 | | −0.518 |
| Gastric contents [b] | 0.67 ± 1.08 | 4.45 ± 9.23 | | −0.585 |

[a] *p*-value has been calculated using Mann Whitney U-test. [b] *p*-value has been calculated using independent *t*-test.

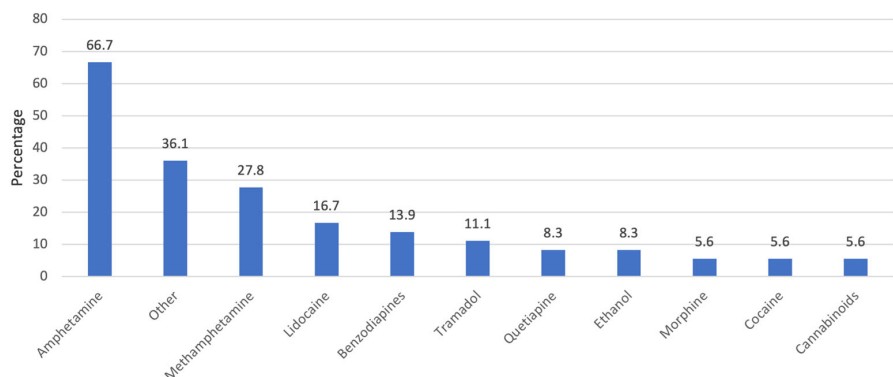

**Figure 2.** Other drugs detected in the studied cases.

### 3.7. History of Drug Abuse

The THC metabolic elements and the history of drug abuse did not show a significant difference (all *p* > 0.05). The average values for THC, THC-OOH, and THC-OH metabolic variables were not significantly impacted by the history of drug abuse (as seen in Table 8).

### 3.8. Body Mass Index and THC Metabolites

Table 9 presents the results of the analysis conducted on the relationship between BMI levels and THC metabolic parameters. The findings indicated no significant difference between the BMI levels and the THC metabolic parameters with the exception of THC-OH concentration in urine samples. The statistical analysis revealed that the relationship between THC-OOH metabolic parameters and BMI was not significant (all *p* > 0.05). Nonetheless, a significantly lower average value of THC-OH in urine was observed in individuals who had a normal BMI and passed away (F = 5.087; *p* = 0.012). However, no other THC-OH metabolic parameters were found to have a significant correlation with BMI.

**Table 9.** Comparison of THC and its metabolites in relation to patients' BMI.

| Specimens | Level of BMI | | | F-Test | *p*-Value |
|---|---|---|---|---|---|
| | Metabolites | | | | |
| | Normal<br>Mean ± SD | Overweight<br>Mean ± SD | Obese<br>Mean ± SD | | |
| | Δ9—Tetrahydrocannabinol | | | | |
| Blood [a] | 17.0 ± 62.0 | 12.3 ± 26.3 | 3.97 ± 1.85 | 0.205 | 0.463 |
| Urine [a] | 2.36 ± 2.74 | 8.41 ± 14.1 | 2.75 ± 2.66 | 2.081 | 0.248 |
| Liver [b] | 0.88 ± 0.66 | 2.18 ± 1.16 | 1.87 ± 2.19 | 1.994 | 0.179 |
| VH [b] | 0.67 ± 0.74 | 1.14 ± 0.84 | 1.22 ± 0.74 | 1.516 | 0.237 |

**Table 9.** *Cont.*

| Specimens | Level of BMI | | | F-Test | *p*-Value |
|---|---|---|---|---|---|
| | Metabolites | | | | |
| | **Normal**<br>**Mean ± SD** | **Overweight**<br>**Mean ± SD** | **Obese**<br>**Mean ± SD** | | |
| Kidney [b] | 0.95 ± 0.98 | 5.19 ± 5.99 | 4.14 ± 5.03 | 0.587 | 0.569 |
| Bile [a] | 81.4 ± 100.0 | 88.5 ± 105.2 | 1.95 ± 0 | 0.322 | 0.175 |
| | 11-nor—Δ9-Tetrahydrocannabinol-9-Carboxy acid | | | | |
| Blood [b] | 18.2 ± 16.7 | 20.3 ± 18.9 | 8.48 ± 7.63 | 1.253 | 0.298 |
| Urine [a] | 285.3 ± 313.8 | 673.8 ± 1052.2 | 138.9 ± 177.7 | 1.941 | 0.219 |
| Liver [a] | 64.5 ± 96.2 | 110.5 ± 169.9 | 1.04 ± 1.27 | 0.530 | 0.389 |
| VH [b] | 0.74 ± 0.66 | 0.96 ± 0.51 | 1.03 ± 0.44 | 0.755 | 0.479 |
| Kidney [b] | 115.9 ± 216.0 | 160.3 ± 349.3 | 2.03 ± 2.85 | 0.236 | 0.794 |
| Bile [a] | 134.2 ± 357.1 | 28.9 ± 39.9 | 2.09 ± 1.09 | 0.554 | 0.113 |
| | 11-Hydroxy-Δ9-Tetrahydrocannabinol | | | | |
| Blood [b] | 2.24 ± 7.48 | 1.12 ± 0.96 | 0.81 ± 0.57 | 0.269 | 0.766 |
| Urine [b] | 0.33 ± 0.43 | 0.94 ± 0.72 | 0.82 ± 0.44 | 5.087 | 0.012 ** |
| Liver [b] | 0.48 ± 0.36 | 1.04 ± 0.75 | 0.63 ± 0.88 | 1.186 | 0.339 |
| VH [b] | 0.24 ± 0.39 | 0.58 ± 0.45 | 0.63 ± 0.42 | 2.778 | 0.079 |
| Kidney [b] | 0.55 ± 0.47 | 0.71 ± 0.40 | 0.56 ± 0.78 | 0.172 | 0.844 |
| Gastric contents [b] | 5.62 ± 12.0 | 2.42 ± 4.33 | 0.46 ± 0.65 | 0.475 | 0.630 |

[a] *p*-value was calculated using Kruskal–Wallis test. [b] *p*-value was calculated using One-Way ANOVA; ** Significant at $p < 0.05$ level.

## 4. Discussion

In this study, various specimens such as BNaF, urine, liver, vitreous humor, kidney, bile, brain, and gastric contents were examined using LC-MS/MS to determine the presence of THC and its metabolites. The analysis method used in this study was shown to be reliable through experimentation, demonstrating its ability to accurately analyze post-mortem samples from autopsy tissues submitted to our laboratory.

Blood is a commonly used specimen for forensic toxicology investigations, in some cases, blood is not available due to the nature of fatal accidents [8,15]. However, this was not the case in this study, as most of the cases were for routine postmortem toxicology analysis, compared to other studies that discussed pilots or aviation accidents. In this study, BNaF cases were positive for THC in 86% of available blood samples. 100% of blood specimens were positive for THC-COOH while THC-OH was positive in only 23 specimens. The proportion of BNaF specimens that tested positive for the level of THC, THC-COOH, and THC-OH was found to be higher than earlier reports [9,11].

This study revealed that urine, a non-blood bodily fluid, could be utilized for the analysis of cannabinoid metabolites [21]. Despite examining almost four times more cases than the Saenz et al. [13] study, a lower proportion of urine specimens tested positive for THC-OH and THC. Both studies employed comparable techniques to preserve urine samples; however, the present study used alkaline hydrolysis, while Saenz et al. [13] used enzymatic hydrolysis. Saenz et al.'s [13] report showed that 70% of the urine specimens tested positive for THC-COOH and THC-OH, with concentrations ranging from 24.2 to 970 ng/mL and 11.7 to 620 ng/mL, respectively. This can be explained by the efficiency of enzymatic hydrolysis in deconjugating the THC-glucuronide and THC-OH-glucuronide to their free form, compared to alkaline hydrolysis [9,22].

The concentration of THC and THC-COOH in bile specimens in the current study was found to be higher compared to previous reports [3,8,11], which reported a concentration range of 0.75–50.4 ng/mL and 201–307 ng/mL, respectively. The case is different with THC-OH, the concentration was lower than previously reported, in these reports, the concentrations ranged from 0.98 to 230 ng/mL [3,8,11]. In contrast, our findings aligned with the Gronewold and Skopp study [16] which reported THC-OH concentrations ranging

from 1.0 to 54 ng/mL. All 17 cases tested positive for THC-COOH, while THC and THC-OH were negative in five and four out of 17 cases, respectively. This suggests that bile can be a useful specimen for post-mortem analysis, complementing blood. Bile, like urine, is a bodily excretion and therefore may have a higher concentration of cannabinoids and related compounds due to the presence of THC and THC-OH in their free form. In contrast, the concentration of drugs and their conjugates in blood and other specimens is poorly correlated. This is why the concentration of cannabinoid metabolites in bile specimens was found to be higher than in blood specimens [11]. In vitreous humor specimens, the results of our study were similar to previous reports that stated that vitreous humor is not suitable for testing THC and its metabolites, in fact, this can be supported by a few studies that identified THC and its metabolites in vitreous humor samples [2,9,23]. In previous studies, THC concentration was found in the range of 1–8 ng/mL in VH without hydrolysis [2,9]. However, even after using enzymatic hydrolysis, by Saenz et al. [13], VH specimens tested negative for THC. The reason could be the small sample size of specimens since only two VH cases were included in Saenz et al. study [11]. THC-COOH has a better detectability rate than THC-OH, THC-COOH in vitreous humor as was reported in two previous postmortem studies with concentrations ranging from 0.6 to 8 ng/mL [2,9]. THC-OH was detected in three cases in the current study but had not been reported in any previous reports. Petterson et al. [24] referred the low detectability of THC and their metabolites in vitreous humor to the nature of the specimens, known to be high in protein-binding and lipophilicity which makes the detection of THC in aqueous environments such as vitreous humor difficult.

From the previous reports, only two previous studies reported THC and its metabolites in gastric contents obtained from postmortem cases [9,13]. Gronewold and Skopp reported two positive stomach contents cases for THC and the concentration range for THC-COOH, THC-OH and THC were found to be 63–2,440 ng/mL, 7–760 ng/mL, and 2–130 ng/mL, respectively [13]. In an Al-Asmari et al. study [9], higher levels of THC were found than THC-COOH in most of the gastric contents samples, which is in agreement with the current study. The higher THC concentrations in gastric contents can be explained by the role of the route of administration, as there is direct contact with the stomach contents and the smoked cannabinoid [24]. The second reason for this high THC concentration in the stomach content can be related to the enterohepatic passage of cannabinoids during metabolism. The elimination can lead to a build-up of THC-glucuronide in the gastrointestinal tract where they are de-conjugated to free THC by the hydrolytic enzymes [25].

In previous studies, both liver and kidney specimens were found to be positive for all cannabinoid metabolites. In the Kemp et al. [15] and the Gronewold and Skopp [13] studies, THC and THC-COOH were tested in liver and kidney specimens and found positive in most cases. Similar to the current investigation, THC-COOH was found to be positive in both matrices which is in agreement with previous reports [11,13–15]. In these studies, THC-COOH concentrations ranged between 8.0 ng/g and 3894 ng/g. In agreement with our study, THC in the liver tested positive in Saenz et al. [11], Kemp et al. [15] and Al-Asmari [14] reports, in these reports, THC was found positive in two cases (range: 22.3–52.2 ng/g), in eight cases (range: 23–237 ng/g) and in one case (25 ng/g), respectively. In fact, THC-OH was rarely reported in liver tissues, it was reported by Saenz et al. [11] in 5 out of 11 cases (range: 1.27–66.1 ng/g) and 4 out of 5 cases as reported by Gronewold and Skopp [16] (range: 1.6–4.1 ng/g). Similarly, all kidney specimens were positive for THC-COOH (range: 3–1774 ng/mL), with THC found in the range of 1–450 ng/mL, and only two studies reported positive kidney specimens for THC-OH that ranged from 1.3 to 135 ng/g [11,14]. The high concentration of THC-COOH in kidney specimens can be due to the role played by the kidney in eliminating cannabinoids. This kind of hydrolysis procedure increased the probability of the detection of THC and THC-OH when enzymatic hydrolysis is used compared to that of alkaline hydrolysis. To support this hypothesis, Gronewold and Skopp [13] reported a direct determination of analytes of interest without hydrolysis, in that study, THC and THC-OH, THC-COOH and THC-COOH-glucuronide

were included in the method of analysis, THC-OH-glucuronide and THC-glucuronide were not included in the analysis as they were not commercially available. A higher concentration of THC-COOH-glucuronide was obtained, while THC and THC-OH were not detected or only detected in trace amounts. This indicates that both THC and THC-OH are conjugated extensively in a similar manner to that of THC-COOH and required an appropriate method of hydrolysis to cleave THC-OH and THC [14]. In contrast, alkaline hydrolysis used by other studies was able to detect THC and THC-OH metabolites in liver and kidney specimens but with very low concentrations [14,15]. The use of enzymatic hydrolysis was found to enhance the detectability of THC-THC-OH in liver and kidneys specimens as reported by Saenz et al. [11]. In some cases, conjugated metabolites are found to be cleaved to their free form if not stored properly. This occurs especially in internal solid tissue, such as the liver and kidney as has been seen with morphine glucuronide [26].

In this study, THC, THC-OH, and THC-COOH were detected in the brain. This is in agreement with previous studies where THC, THC-OH, and THC-COOH were detected in most of the brain specimens, though in low concentrations [7,11,13]. THC-COOH was detected only in brain specimens in Kemp et al. studies [15]. The reason could be the small number of cases (four brain cases) in our study. It is well known that brain tissues could be an ideal alternative to blood. The brain is reported to be highly perfused, making it easier for THC to cross the blood-brain barrier, and therefore cannabinoid metabolites were found in the brain but not in the blood. The nature of brain THC active metabolites can be determined in brain samples even when it is reported as negative in blood samples [14,27].

To the best of the authors' knowledge, this is the first study to report THC-OH in vitreous humor samples. In our study, no statistical significance was found between THC-COOH and THC metabolites and BMI for all specimens analyzed. However, the mean value of THC-OH in urine was significantly lower in individuals with a normal BMI. Hence, higher levels of THC-OH in urine could be observed in individuals with a high BMI compared to other bodily fluids or tissues. The results also indicated that the distribution of THC and its metabolites can be affected by renal function, BMI/body composition, and gender, but not by prior use of other drugs.

**5. Conclusions**

We confirmed that the LC-MS/MS is a reliable approach for analyzing THC and its metabolites in post-mortem sample investigations. The use of multiple specimens in post-mortem analysis can improve the accuracy of cannabinoid investigation. In addition, all types of specimens were found to be suitable for testing cannabinoid metabolites, except for vitreous humor which showed a low rate of cannabinoid metabolite detectability. However, to the best of the authors' knowledge, this is the first study to report THC-OH in vitreous humor samples. The latter matrix is promising in cases where blood is unavailable. The conclusion of the study was that the distribution of THC and its metabolites may be influenced by BMI/body composition but not by a history of other drug use. Although blood samples are the standard choice for the analysis of cannabinoids, the inclusion of other bodily fluids or tissue specimens should be considered for two reasons: as complementary specimens to blood and to provide more information in postmortem toxicology investigations, in cases where no blood samples are available, which enhances the quality of the investigation. To the best of the authors' knowledge, this is the first study to correlate THC metabolites concentration and body mass index and history of drug abuse.

**Supplementary Materials:** The following supporting information can be downloaded at: https://www.mdpi.com/article/10.3390/forensicsci3020025/s1, Table S1: Linear Coefficient determination, LODs and LLO ng/ml and upper limit of quantification for all specimens of interest in the current study (*n* = 5); Table S2: Method validation parameters for the analysis pf Δ9-tetrahydrocannabinoland its metabolites using Liquid chromatography tandem mass spectrometry; Table S3: Demography of 43 postmortem cases included in this work.

**Author Contributions:** Conceptualization, T.A.Z., L.A.-Q. and H.A.; methodology, A.I.A.-A., A.E.A., H.A. and T.A.Z.; software, A.E.A., T.A.Z. and H.A.; validation, A.E.A., R.A.-M., A.A.-G., M.A., F.A., L.A.-Q., A.A.M., J.B., S.T., A.A., S.A.-G. and H.A.; formal analysis, A.I.A.-A., A.A.-G., R.A.-M., M.A., F.A., L.A.-Q., A.A.M., S.T., A.A. and S.A.-G.; investigation, T.A.Z., A.I.A.-A.; resources, A.E.A. and H.A.; data curation, A.I.A.-A., A.A.-G., A.E.A., M.A.H. and H.A.; writing—original draft preparation, A.I.A.-A., L.A.-Q. and T.A.Z.; writing—review and editing, T.A.Z., M.R.P. and J.B.; visualization, A.I.A.-A. and T.A.Z.; supervision, A.I.A.-A. and T.A.Z.; project administration, T.A.Z. and H.A. All authors have read and agreed to the published version of the manuscript.

**Funding:** This research received no external funding.

**Institutional Review Board Statement:** The study was conducted in accordance with the Declaration of Helsinki and approved by the Ethics Committee of Jeddah Health Affair, Ministry of Health in Saudi Arabia, research code: ethical approval no. H-02-J-002).

**Informed Consent Statement:** Not applicable.

**Data Availability Statement:** The data underlying this article can be requested from the corresponding author.

**Acknowledgments:** The authors would like to thank all the staff at the Forensic Toxicology Department-Jeddah Poison Control and Forensic Medical Chemistry Center for supporting this work.

**Conflicts of Interest:** The authors declare no conflict of interest.

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
