# Peer review of "Comparison between Blood, Non-Blood Fluids and Tissue Specimens for the Analysis of Cannabinoid Metabolites in Cannabis-Related Post-Mortem Cases"

_forensicsci, doi:10.3390/forensicsci3020025_

Round 1

Reviewer 1 Report

   1) Title: the title of this study in about "Cannabis related post-mortem cases” but we don’t know which is the cause of death. I’ts unknown? (car accident, acute poisoning, sudden unexpected death), please explain

   2) how long after death samples were collected? What about putrefaction?   

   3) Out of 28 references 3 are by Al-asmari, if he’s the same author it’s recommended to add more references to support it

   4)    From line 90 to 92 there is a repetition. Maybe you should postpone the paragraph about metabolites (line 72-89) 

   5)    Line 149 BNaF spell out acronym

   6)    Line 165 spell out acronym 

   7)    Line 160 4 brain tissue e blood liver infiltrations around liver samples?how many of the first and how many of the second one? Please specify  

   8)    Table 1 should be under line 203

   9)    Line 215 “amphetamines”?please explain

  10)  Line 230 “Prevalence of the deceased with a 228 previous history of recreational drug use was 18.6%, while the prevalence of the 229 deceased with other drug use was 83.7%”. Does the information about other drug use come from history or analysis?please explain

  11)  Line 288 space missing after dot and A

  12)  Line 289 “with”?please explain

  13)  Line 296 e 297 erase space

  14)  Line 309 S-content spell out acronym

  15)  Line 311 space missing after dot and A

  16)  Line 334 erase line

  17)  Line 339 “the role of other drugs detected” are they involved in the cause of death?please explain

  18)  Table 7 what’s “other” on the second bar?please explain

  19)  Line 438 why the authors investigate the correlation between BMI ang THC metabolites? Please explain also in the conclusion

  20)  Line 489 The role of urinay matrix is widely established in the scientific literature. Please add some references.

  21)  Line 540 erase space after “[18]”

  22)  Line 603 the authors state that “bmi influence the distribution”. On line 585 the authors state that there’s no statistical significance. Please explain

Author Response

Dear Reviewer,

Thank you so much for your comments and suggestions. Please find attached a document with a table showing your comments and our responses. We hope we answered and corrected the matters to your satisfaction.

Reviewer 2 Report

Authors should have clearly mentioned the procedure of collection of tissue samples. They have just mentioned 3 parts in liver. It does not specify. 

Collection of vitreous fluid and urine also must be clearly mentioned. 

Must be improved

Author Response

(The authors gave the same response as above.)

Reviewer 3 Report

The topic is very interesting and this study could be useful in post-mortem interpretation. In my opinion, several amendments are required:

Abstract:

- Line 23: open ( not closed

- Brief description of analytical method must be added (instrumentation, column, kind of extraction, main MRM)

- Main results must be summarized.

Introduction:

- Introduction is too long and should be revised. Statistics about Cannabis consumption must be reduced. 

- Description of previous study can be reduced since they are described in discussion.

Materials and methods

- The authors considered blood from subclavian sites. Considering the PM redistribution, did you evalutate also blood from peripheric sites (i.e. femoral)? And other studies?

- Many abbraviations should be described: BNaF, BN, VH.

- page 4, line 164: please, correct ul (µL?)

- page 4, line 172: 200 µl was repeated twice.

- Chromatograms at LOQ values and/or for a real sample should be added

-Main validation parameters should be reported also in table.

- page 6, line 216: "the matrix effects for amphetamine..."???

- LOQ value was 2.0 ng/mL, but le calibration curve started from 1.0 ng/mL. This is not possible, as below the LOQ you can not be able to quantify a substance.

- The calibration range for solid tissues should be added. 

- how did you build up the blank and calibration samples? Which blank matrix was used?

Author Response

(The authors gave the same response as above.)

Round 2

Reviewer 3 Report

The paper was amended as required.